# Umbilical cord length and neurodevelopmental disorders, a national cohort study

Cathrine Ebbing[1,2]*, Anne Halmoy[3,4], Svein Rasmussen[1], Karen K. Mauland[1], Jørg Kessler[1,2], Dag Moster[5,6]

1 Department of Clinical Science, University of Bergen, Bergen, Norway, 2 Department of Obstetrics and Gynecology, Haukeland University Hospital, Bergen, Norway, 3 Department of Psychiatry, Haukeland University Hospital, Bergen, Norway, 4 Department of Clinical Medicine, University of Bergen, Bergen, Norway, 5 Department of Global Public Health and Primary Care, University of Bergen, Bergen, Norway, 6 Department of Pediatrics, Haukeland University Hospital, Bergen, Norway

* cathrine.ebbing@uib.no, cathrine.ebbing@helse-bergen.no

## Abstract

### Introduction

Adversities in fetal life are known risk factors for neurodevelopmental disorders (NDD). Despite the pivotal role of the umbilical cord, little is known about its associations to later NDD.

### Objective

To estimate the associations between umbilical cord length and NDD (Attention-deficit/hyperactivity disorder (ADHD), autism spectrum disorder (ASD), intellectual disability (ID), cerebral palsy (CP), epilepsy, impaired vision or hearing), and whether associations differed by sex.

### Materials and methods

A prospective population-based cohort study including all liveborn singletons in Norway from 1999, through 2013 and followed up through 2019. Data were retrieved from The Medical Birth Registry of Norway and linked with other national health and administrative registries. Exposures were extreme umbilical cord length (empirical percentile <5th or ≥ 95th percentiles). Main outcome measures were NDD (ADHD, ASD, ID, CP, epilepsy, impaired vision or hearing). Associations with umbilical cord length were assessed using logistic regression.

### Results

The cohort consisted of 858,397 births (51.3% boys). We identified 33,370 persons with ADHD (69.8% boys), 10,818 had ASD (76.0% boys), 5538 ID (61.4% boys), 2152 with CP (59.9% boys), 8233 epilepsy (55.0% boys), 900 impaired vision (boys 55.0%), and 11,441 impaired hearing (boys 52.8%). Cord length was positively

**Data availability statement:** Legal restrictions do not permit the authors to provide the data that constitute the basis of this study. The main data utilized are available from the data owner, the Norwegian Institute of Public Health (https://www.fhi.no/en/more/research--access-to-data/), after obtaining approval from The Regional Committee for Medical Research Ethics (https://rekportalen.no/), for researchers who meet the criteria for access to confidential data. Contact information: The Medical Birth Registry of Norway, University of Bergen, P.O. Box 7804, 5020 Bergen, Norway. Code availability: The data are confidential and cannot be shared.

**Funding:** The project was supported by the Gerda Meyer Nyquist Gulbrandson and Gerdt Meyer Nyquist's Fund, and the Norwegian SIDS and Stillbirth Society https://lub.no/. Prof. Dag Moster was awarded the grants. The funders played no role in conducting the research and writing the paper.

**Competing interests:** The authors have declared that no competing interests exist.

associated with ADHD (OR 1.15; 95%CI 1.09–1.22), i.e., the risk increased with long cord and decreased with short cord, regardless of sex. A short cord was positively associated with ID (OR 2.42; 95%CI 2.17–2.69), impaired hearing (OR 1.41; 95%CI 1.29–1.54), and epilepsy (OR 1.31; 95%CI 1.18–1.46). CP was associated with both short and long cord (OR 1.31; 95% CI 1.07–1.61 and 1.34, 95%CI 1.13–1.60, respectively). There was no association between cord length and impaired vision.

## Conclusions

This first population study finds that umbilical cord length is associated with NDD. The findings support the hypothesis that neurodevelopment and development of the umbilical cord share pathways.

---

## Introduction

Neurodevelopmental disorders (NDD) represent a diverse spectrum of disabilities linked to the functioning of the brain [1]. In the current version of the Diagnostic and Statistical Manual of Mental Disorders (DSM-5), the term NDD includes attention-deficit hyperactivity disorder (ADHD), autism spectrum disorder (ASD), intellectual disability (ID) and specific learning- communication- and motor disorders, including cerebral palsy (CP) [1]. These disorders are usually diagnosed during childhood and persist until adulthood [2]. When broadening the concept to include epilepsy and problems with sensory functions, such as impaired vision or hearing, approximately 15% of children are affected by these disorders, constituting a leading cause of disease-related burden in this group worldwide [2–4].

The etiology of NDD is intricate, involving a composite interplay of genetic, bio-logical, psychosocial, and environmental factors [5,6]. Well established risk factors such as preterm birth and low birth weight [5,7,8] suggest that adverse development is present during fetal life. Boys are more susceptible to NDD [4]. Whether observable sex-specific growth and development patterns from early stages of conception [9–14] are relevant to NDD is however unknown.

What determines the umbilical cord length is largely unknown, but the cord, which develops throughout pregnancy, is longer in boys than girls [12,15]. The cord length may also be influenced by maternal factors, fetal- and placental size, and fetal motor activity, but the latter is disputed [15–17]. Notably, fetal motor activity has been demonstrated to stimulate umbilical cord growth [18,19], whereas congenital conditions associated with neonatal hypotony or immobility (such as chromosomal aberrations) have been associated with a short cord [20,21]. Despite the pivotal role of the umbilical cord, studies on umbilical cord length and later neurodevelopment are scarce. A Finnish institutional study found an association between cord length and childhood epilepsy (a short cord was associated with epilepsy), but population studies and studies on other NDD outcomes are lacking [21]. Long-term consequences of placental size and birthweight on health follow sex-specific patterns [14,22], but whether there is an association between cord length and NDD (other than epilepsy),

and whether these associations are sex-differential, is unknown. Thus, the objectives of this prospective population-based cohort study were to investigate the associations between umbilical cord length and a range of NDD (ADHD, ASD, ID, CP, epilepsy, impaired vision, and hearing) and to explore if associations differ between boys and girls.

## Materials and Methods

We performed a comprehensive national cohort study by including all singleton live births in Norway between 1999–2013 with gestational age between 22 and 44 weeks. (When gestational age was missing, births with birthweight 500 grams or more were included.) Data were sourced from the Medical Birth Registry of Norway (MBRN). Gestational age was based on ultrasound (97%) or the last menstrual period if ultrasound information was missing. The cohort was followed through 2019 by linkage to other national databases. Outcome information (NDD: ADHD, ASD, ID, CP, epilepsy, and impaired vision or hearing) was provided from the National Insurance Scheme (NIS) and the National Patient Registry (NPR). A person was classified as having the outcome if the diagnosis was recorded at least once by the NIS and/or at least twice by the NPR (ICD-9 and -10 codes with the diagnoses can be found in the S1 Supplement). Statistics Norway provided information on maternal and paternal immigrant status and level of education. The unique personal Norwegian identification number allowed linkage of individual data from MBRN with these compulsory registries. Information from all registries was updated through 2019. Since MBRN changed the way it recorded conditions of placenta and umbilical cord in 1998, we included birth years from 1999 onwards to obtain uniform recordings. Given the nature of the diagnoses studied here, namely ADHD, ASD, ID, CP, epilepsy and impaired vision or hearing, and their inability to be diagnosed at birth, the study included only birth years through 2013, allowing a follow-up time ranging from 6 to 21 years.

### Exposures

Exposures were short (<5th percentile) and long (≥95th percentile) umbilical cord. The cord length percentiles were empirical, i.e., constructed based on the distribution of umbilical cord length by parity, sex and gestational age in the study population. Since cord length increases with fetal size, we additionally constructed birthweight-specific percentiles for cord length. All data in the MBRN are registered prospectively. The attending midwife or physician conducted the examinations of the afterbirth and the neonate and entered the requested information into the registration form shortly after delivery. All neonates were examined by a physician and any malformation was noted. The umbilical cord was measured with a disposable tape and length was reported in centimeters.

### Outcomes

We identified individuals with the NDD outcomes through the NIS and the NPR as previously detailed [23]. In Norway, all residents are compulsory insured through the NIS [24]. Benefit is granted for persons with disabilities independently of wealth or income, and the causing medical conditions are recorded. We identified all persons with the NDD diagnoses (See S1 Supplement for ICD- 9 and -10 codes) as the medical reason for the benefit granted. Furthermore, we also received information on the diagnoses from NPR. This register holds individual data on visits to the specialist care in Norway [25]. In our study, a person was registered with the diagnoses if the person received a benefit, or a disability pension based on the diagnoses in the NIS or was registered with at least one of the diagnoses at least twice in the NPR.

### Covariates

The variables included in the models were selected based on their potential impact on the risk estimates; year of birth, parity, maternal and paternal age and educational level, maternal smoking at the beginning of pregnancy, conception by assisted reproductive technology (ART), maternal body mass index (BMI), and maternal medical conditions (any one of asthma, urinary tract infection, hypertension, rheumatoid arthritis, cardiac disease, epilepsy, thyroid disease, pre gestational diabetes mellitus). The marital status of the mother, maternal and paternal level of education (categorized into five

levels) and immigrant status were included. A child was considered to have immigrant parents if both parents were born abroad. Statistics Norway provided information on level of education and immigrant status of parents, while MBRN provided information on the other covariates. The study adhered to the Strengthening the Reporting of Observational Studies in Epidemiology (STROBE) guidelines [26], and The Regional Committee for Medical and Health Research Ethics in Norway and the Registry Owners approved this study and waived the need of informed consent for the use of these registry data (REK 2018/1789).

## Statistical analysis

To analyse the association between umbilical cord length and the NDD outcomes, we used multilevel logistic regression which allowed for the within-mother clustering (>1 births in the same woman), with adjustments for possible confounding variables to calculate odds ratios (ORs) and 95% confidence intervals (95% CI). In additional analyses, we stratified by sex of the child, and gestational age at birth (<37 and ≥ 37-gestational weeks). We used a directed acyclic graph to inform decisions as to which variables to include in the models (Fig 1).

Furthermore, to assess if the associations between the main exposures (long and short umbilical cord) and NDD persist with unmeasured confounding, we used Bayesian simulation analysis with standard normal informative priors [27]. We used a simple logistic regression model (fixed effect: $\beta 0 + \beta 1$ exposure (0 or 1)) in the Bayesian simulation where $\beta 0$ and $\beta 1$ are constants. We assumed that adding an influential, unmeasured confounder in the regression would neutralize the associations between the exposures and NDD, decreasing the regression coefficient ($\beta 1$; standard deviation, sd) for the exposure to 0; sd, corresponding to an odds ratio of 1; 95% CI: $\exp(0 \pm sd\ 1.96)$. The sd was set equal to that generated by a model without the assumption (S1 Table).

We used Statistical Package for the Social Sciences for Windows (version 29; SPSS, Chicago IL, USA) and Stata version 18.5 for the statistical analyses.

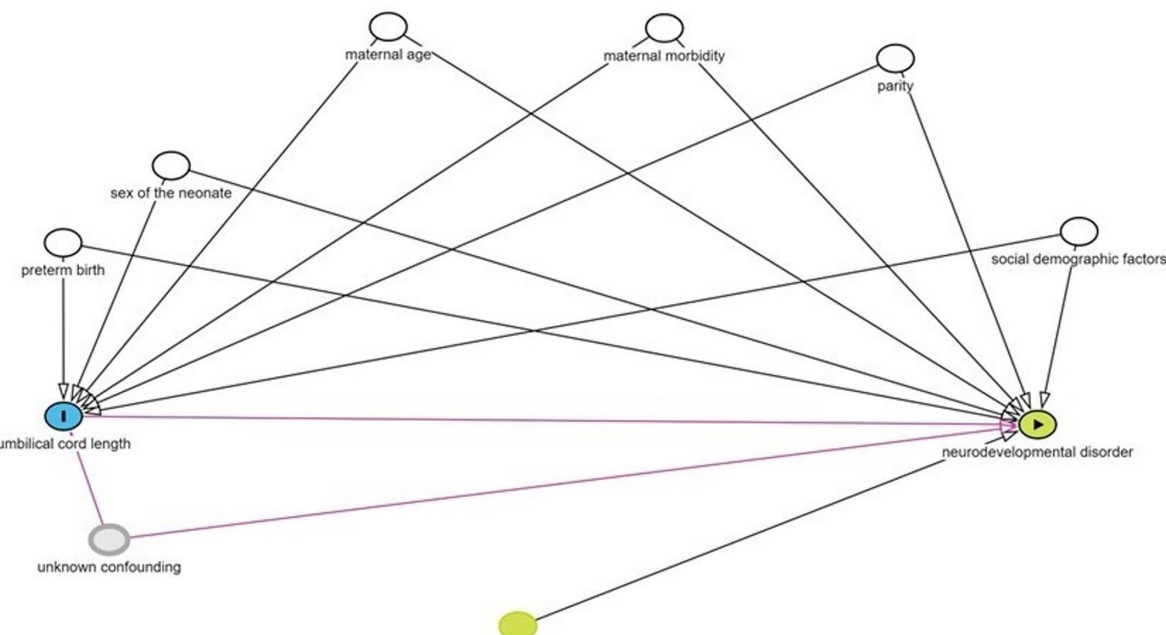

**Fig 1. Directed acyclic graph (DAG) describing the relationship between pregnancy, maternal and paternal factors, umbilical cord length and neurodevelopmental disorder in the child.** White nodules represent covariates that are included in the model. Social demographic factors contain maternal and paternal education level, marital status of the mother, immigration status of the child. Grey nodule; Unknown confounding factors, e.g., genetic confounding.

## Results

In the study population of 858,397 individuals, we identified 33,370 persons with ADHD (70% boys), 10,818 with ASD (76% boys), 5538 with ID (61% boys), 2152 with CP (60% boys), 8233 with epilepsy (55% boys), 11,441 with impaired hearing (53% boys) and 900 with impaired vision (55% boys). Cord length percentiles were based on 795,489 births with known gestational age and cord length. Characteristics of the study population are shown in Table 1.

The occurrences of ADHD, ASD, ID, CP, epilepsy, impaired hearing and vision declined by year of birth, especially for ADHD (Fig 2). The association of cord length with ADHD was positive and exhibited a pattern where the risk increased with longer and decreased with shorter cords (Figs 3 and 4): This was found irrespective of sex. ASD was associated with long cord, especially in boys, in analyses stratified by sex (Table 2).

Conversely, ID, impaired hearing, and epilepsy were positively associated with short cord (Figs 3 and 4 and Table 2). The link between ID and short cord was consistently observed, regardless of the individual's sex. Moreover, boys with short cord exhibited reduced odds of ADHD (Table 2). CP was associated with both long and short cord, and overall, umbilical cord length showed no association with impaired vision (Figs 3 and 4 and Table 2).

The associations between cord length and the NDD did not differ notably between boys and girls except that long cord was positively associated with ASD and CP in boys only (Table 2).

Since cord length correlates with fetal size, we also used birthweight-specific percentiles for cord length; When using birthweight-specific percentiles for cord length, the reported associations were largely unaltered, but with wider confidence intervals (S2 Table).

Co-occurrence of conditions was frequently observed within our study population. Malformations were documented in 4.4% of the total population (37,546 out of 858,397) and were more prevalent in individuals diagnosed with any of the studied NDD (In 19.5% of persons with ID, 18.4% with CP, 13.4% with impaired hearing, 12.4% with impaired vision, 9.1% with epilepsy, 7.4% with ASD and 5.4% with ADHD).

We stratified the population based on gestational age at birth; the associations remained between short cords and ID, impaired hearing, and epilepsy in both term and preterm born, and long cord was associated with ADHD, ASD and CP in term births (Table 3).

We included year of birth, maternal age and parity as co-variates in the regression analyses. Adding paternal age, immigration status or any maternal medical condition (any one of the following: asthma, chronic hypertension, kidney disease, rheumatoid artritis, epilepsy, pre-gestational diabetes mellitus, or thyroid disease) into the models in addition to year of birth attenuated the associations. Our sensitivity analyses indicated that our main results were not explained by variables not included in our database. After including the assumption of an unknown confounder in the regression analyses, the associations persisted (S1 Table).

Since maternal BMI before pregnancy was available only in a subset of the population, we repeated the analyses (with NDD as outcomes) restricted to cases with information on maternal BMI (n = 174,692). We found no effect of adjusting for maternal BMI on the risk estimates. Likewise, no effect was observed on the risk estimates when we adjusted for ART.

## Discussion

In this national cohort study, we found associations between umbilical cord length and NDD diagnosed during childhood/adolescence. The findings were most pronounced in term births, and although the odds were moderately increased, the risk estimates seemed robust to potential unmeasured confounding. Some findings showed a dose-response pattern, supporting the biologic plausibility of the associations.

This study is, as far as we know, the first to evaluate the association of cord length to a variety of NDD outcomes, on a population level. Our findings support the hypothesis that umbilical cord length might reflect fetal motor activity and hence fetal neurobehavioral development. The association between long cord and ADHD is in line with the notion that

**Table 1. Cohort characteristics of singleton births in Norway (1999–2013)[a].**

| [a]Characteristic | Total population (n (%)) | [d]Long cord (n (%)) | [d]Short cord (n (%)) |
|---|---|---|---|
| **Maternal age (years)** | | | |
| <20 | 3262 (0.3) | 190 (0.4) | 98 (0.3) |
| 20–24 | 14317 (1.7) | 786 (1.7) | 434 (1.4) |
| 25–29 | 117726 (14.7) | 6377 (14.3) | 4209 (13.6) |
| 30–34 | 258877 (32.5) | 13709 (30.8) | 10117 (32.8) |
| 34–39 | 262925 (33.0) | 14617 (32.8 | 10560 (34.2) |
| 40+ | 138381 (17.3) | 8771 (19.7) | 5388 (17.4) |
| unknown | 1 (0) | 0 | 0 |
| **Parity** | | | |
| 0 | 330827 (41.5) | 17891 (40.2) | 11984 (38.9) |
| 1 | 284933 (35.8) | 14410 (32.4) | 12689 (41.1) |
| 2 | 127186 (15.9) | 8255 (18.5) | 4482 (14.5) |
| 3 | 35635 (4.4) | 2648 (5.9) | 1103 (3.5) |
| 4+ | 16908 (2.1) | 1246 (2.8) | 548 (1.7 |
| **Marital status** | | | |
| Not single | 733819 (92.2) | 40943 (92.1) | 28546 (92.6) |
| Other | 61670 (7.7) | 3507 (7.8) | 2260 (7.3) |
| **Maternal education (years)** | | | |
| <8 | 8071 (1.0) | 398 (0.8) | 392 (1.2) |
| 8–10 | 113238 (14.2) | 6381 (14.3) | 4550 (14.7) |
| 11–12 | 19798 (2.4) | 1189 (2.6) | 785 (2.5) |
| 13–17 | 534189 (67.1) | 30182 (67.9) | 20281 (65.8) |
| 18+ | 104821 (13.1) | 5554 (12.4) | 4087 (13.2) |
| unknown | 15372 (1.9) | 746 (1.6) | 711 (2.3) |
| **Paternal education (years)** | | | |
| <8 | 6508 (0.8) | 353 (0.7) | 280 (0.9) |
| 8–10 | 124514 (15.6) | 6889 (15.4) | 4925 (15.9) |
| 11–12 | 29955 (3.7) | 1778 (4.0) | 1169 (3.7) |
| 13–17 | 503209 (63.2) | 28543 (64.2) | 19178 (62.2) |
| 18+ | 106466 (13.3) | 5577 (12.5) | 4190 (13.6) |
| unknown | 24837 (3.1) | 1310 (2.9) | 1064 (3.4) |
| **Immigrant** | | | |
| No | 697436 (87.7) | 39542 (88.9) | 26449 (85.8) |
| Yes | 97546 (12.2) | 4885 (10.9) | 4340 (14) |
| Not registered | 507 (0) | 23 (0) | 17 (0) |
| **Smoking at pregnancy start** | | | |
| No | 561131 (70.5) | 31337 (70.4) | 21528 (69.8) |
| Sometimes | 12908 (1.6) | 768 (1.7) | 461 (1.4) |
| Daily | 94005 (11.8) | 5355 (12.0) | 3404 (11.0) |
| Not registered | 127445 (16) | 6990 (15.7) | 5413 (17.5) |
| [b]**Body Mass Index** | | | |
| <18.5 | 7165 (4.1) | 151 (4.0) | 61 (3.8) |
| 18.5–<25 | 106998 (61.2) | 1955 (52.9) | 904 (56.5) |
| 25–<30 | 39251 (22.4) | 887 (24.0) | 360 (22.5) |
| 30+ | 21278 (12.1) | 696 (18.8) | 273 (17) |

*(Continued)*

**Table 1.** (Continued)

| ᵃCharacteristic | Total population (n (%)) | ᵈLong cord (n (%)) | ᵈShort cord (n (%)) |
|---|---|---|---|
| ᶜART | | | |
| No | 779277 (97.9) | 43507 (97.8) | 30090 (97.6) |
| Yes | 16212 (2.0) | 943 (2.1) | 716 (2.3) |
| Gestational age at birth (weeks) | | | |
| <27 | 1804 (0.2) | 100 (0.2) | 67 (0.2) |
| 28–32 | 5948 (0.7) | 330 (0.7) | 237 (0.7) |
| 33–36 | 33147 (4.1) | 1993 (4.4) | 1333 (4.3) |
| 37–41 | 701311 (88.1) | 39287 (88.3) | 27048 (87.8) |
| 42+ | 53279 (6.6) | 2740 (6.1) | 2121 (6.8) |
| Sex of the child | | | |
| Boy | 408711 (51.3) | 26071 (58.6) | 13312 (43.2) |
| Girl | 386778 (48.6) | 18379 (41.3) | 17494 (56.7) |
| Congenital malformations | | | |
| No | 760520 (95.6) | 42607 (95.8) | 28841 (93.6) |
| Yes | 34969 (4.3) | 1843 (4.1) | 1965 (6.3) |

ᵃData from the Norwegian Medical Birth Registry linked with other administrative and medical registers

ᵇBody mass index was only registered in a subset of the population (n = 174,692)

ᶜART; assisted reproductive technology.

ᵈEmpirical percentiles for the umbilical cord length based on the study population, customized for parity, based on the total of 795,489 births with known umbilical cord length and gestational age at birth. Short cord; < 5th percentile (ref.: ≥ 5th percentile), Long cord: ≥ 95th percentile (ref.: <95 percentile).

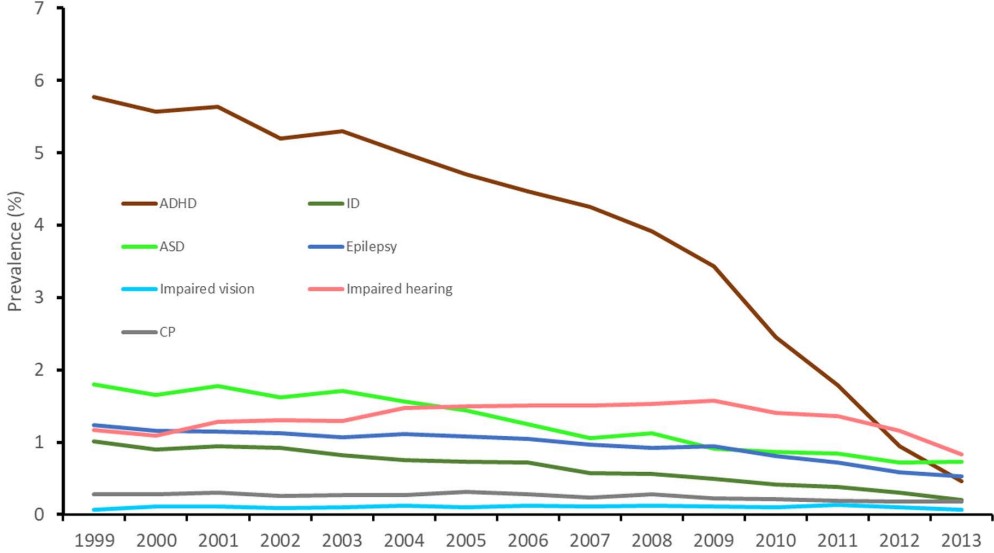

**Fig 2. The prevalence of neurodevelopmental disorders according to year of birth in Norway.**

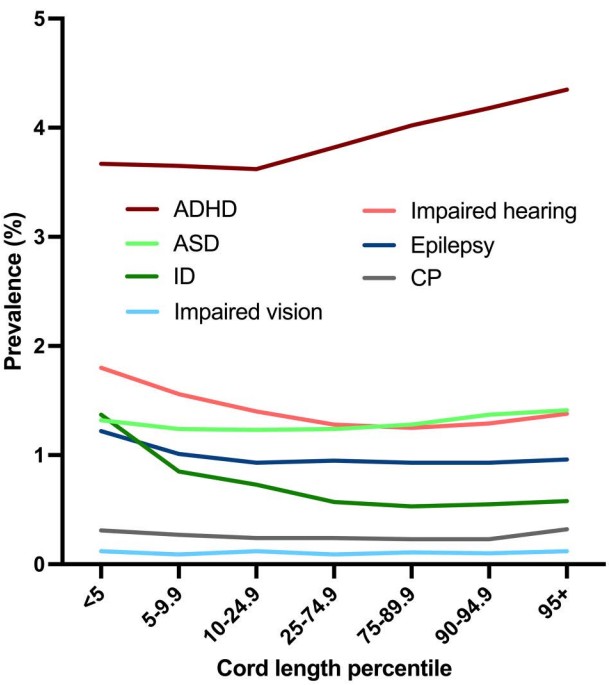

**Fig 3. The prevalence of neurodevelopmental disorders according to cord length percentile.**

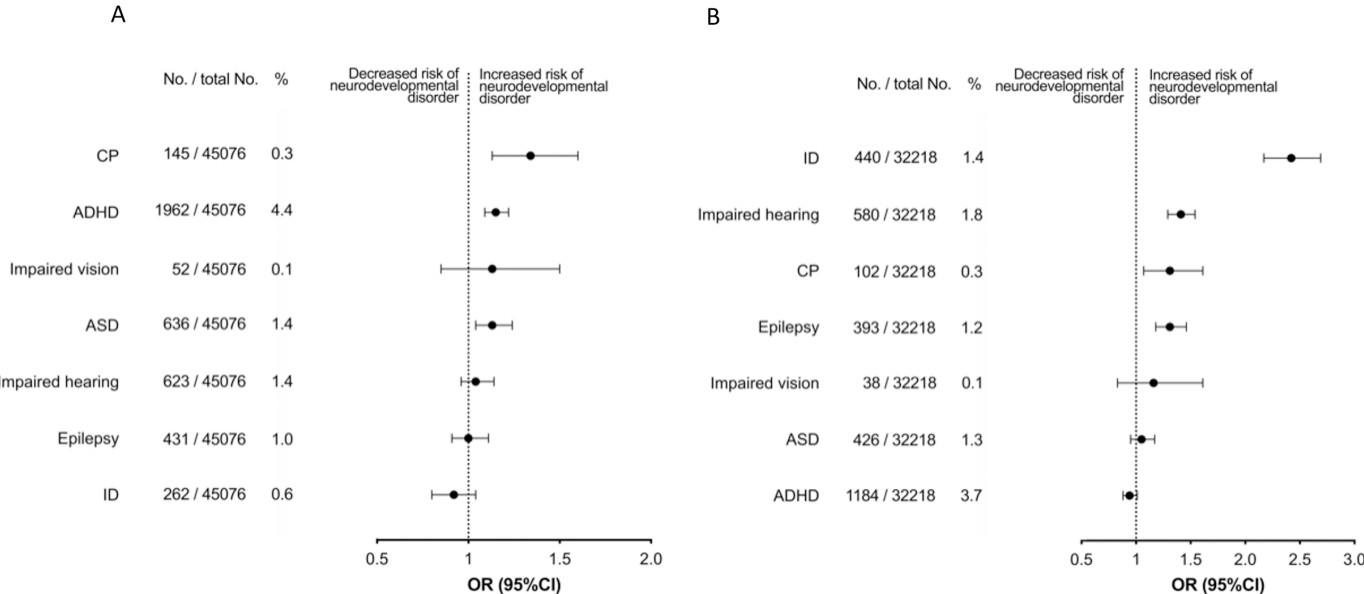

**Fig 4. Long or short umbilical cord (≥95th or<5th percentile) and association to neurodevelopmental disorders among singletons.** Panel A; exposure Long Cord, panel B; exposure Short Cord. CP; Cerebral Palsy, ADHD; Attention-deficit/hyperactivity disorder, ASD; Autism Spectrum Disorder; ID; Intellectual Disability, OR; Odds Ratio, CI; Confidence Interval.

**Table 2. Associations between cord length and neurodevelopmental disorders according to sex of the neonate[a].**

| Sex of the child | | Boy | | Girl | |
|---|---|---|---|---|---|
| Exposure | Outcome | Crude OR (95%CI) | Adjusted OR (95% CI) | Crude OR (95%CI) | Adjusted OR (95% CI) |
| **Long cord** | | | | | |
| | ADHD | 1.18 (1.10-1.26) | 1.16 (1.08-1.24) | 1.15 (1.04-1.26) | 1.10 (1.00-1.22) |
| | ASD | 1.22 (1.10-1.35) | 1.23 (1.11-1.36) | 0.91 (0.75-1.10) | 0.89 (0.73-1.08) |
| | ID | 0.91 (0.77-1.08) | 0.91 (0.77-1.07) | 0.91 (0.74-1.13) | 0.89 (0.72-1.09) |
| | Impaired vision | 0.98 (0.66-1.47) | 0.97 (0.65-1.45) | 1.31 (0.89-1.94) | 1.30 (0.88-1.93) |
| | Impaired hearing | 1.03 (0.91-1.16) | 1.02 (0.90-1.15) | 1.06 (0.93-1.20) | 1.05 (0.92-1.19) |
| | Epilepsy | 0.98 (0.85-1.12) | 0.97 (0.85-1.12) | 1.03 (0.89-1.20) | 1.01 (0.88-1.17) |
| | CP | 1.30 (1.01-1.68) | 1.46 (1.19-1.81) | 1.13 (0.84-1.51) | 1.09 (0.81-1.46) |
| **Short cord** | | | | | |
| | ADHD | 0.92 (0.84-1.00) | 0.97 (0.89-1.06) | 0.98 (0.87-1.10) | 1.03 (0.91-1.16) |
| | ASD | 1.07 (0.90-1.21) | 1.08 (0.95-1.22) | 0.99 (0.79-1.23) | 1.00 (0.80-1.24) |
| | ID | 2.40 (2.08-2.77) | 2.38 (2.07-2.75) | 2.52 (2.13-2.99) | 2.51 (2.12-2.97) |
| | Impaired vision | 1.26 (0.83-1.92) | 1.28 (0.84-1.95) | 1.02 (0.61-1.70) | 1.02 (0.61-1.71) |
| | Impaired hearing | 1.38 (1.22-1.57) | 1.40 (1.24-1.59) | 1.45 (1.27-1.65) | 1.46 (1.28-1.67) |
| | Epilepsy | 1.35 (1.17-1.56) | 1.36 (1.18-1.57) | 1.25 (1.07-1.47) | 1.27 (1.08-1.49) |
| | CP | 1.30 (1.68-1.68) | 1.32 (1.02-1.70) | 1.31 (0.95-1.81) | 1.34 (0.97-1.84) |

ADHD; Attention-deficit/hyperactivity disorder. ASD; Autism spectrum disorder. ID; Intellectual disability. CP; Cerebral palsy.

Short cord; <5th percentile (ref.: ≥ 5th percentile). Long cord ≥95th percentile (ref.: <95 percentile).

[a]Data from the Norwegian Medical Birth Registry and linked population registers 1999–2013 with follow up through 2019.

**Table 3. Associations between cord length and neurodevelopmental disorders according to gestational age at birth (term/ preterm)[a].**

| Gestational age at birth | | Term | Preterm |
|---|---|---|---|
| Exposure | Outcome | Crude OR (95%CI) | Crude OR (95% CI) |
| **Long cord** | | | |
| | ADHD | 1.18 (1.11-1.24) | 0.88 (0.69-1.10) |
| | ASD | 1.15 (1.05-1.26) | 0.89 (0.62-1.27) |
| | ID | 0.94 (0.81-1.07) | 0.76 (0.51-1.13) |
| | Impaired vision | 1.18 (0.87-1.58) | 0.67 (0.20-2.10) |
| | Impaired hearing | 1.06 (0.96-1.15) | 0.91 (0.67-1.22) |
| | Epilepsy | 0.99 (0.89-1.10) | 1.08 (0.78-1.47) |
| | CP | 1.45 (1.19-1.76) | 1.08 (0.75-1.53) |
| **Short cord** | | | |
| | ADHD | 0.93 (0.87-1.00) | 1.06 (0.82-1.35) |
| | ASD | 1.01 (0.9-1.12) | 1.59 (1.13-2.21) |
| | ID | 2.34 (2.08-2.62) | 2.99 (2.22-4.01) |
| | Impaired vision | 1.14 (0.80-1.62) | 1.23 (0.45-3.37) |
| | Impaired hearing | 1.35 (1.22-1.49) | 1.99 (1.53-2.57) |
| | Epilepsy | 1.30 (1.15-1.44) | 1.41 (1.02-1.94) |
| | CP | 1.52 (1.21-1.89) | 0.79 (0.49-1.26) |

ADHD; Attention-deficit/hyperactivity disorder, ASD; Autism spectrum disorder, ID; Intellectual disability, CP; Cerebral palsy, CI; Confidence Interval, OR; Odds Ratio.

Short cord; Short cord (ref.: ≥ 5th percentile), Long cord; Long cord (ref.: <95 percentile).

[a]Data from the Norwegian Medical Birth Registry and linked population registers 1999–2013 with follow up through 2019.

hyperkinesia during fetal life is associated with longer cords ("stretch hypothesis") [19]. Conversely, our study confirmed ID, a group of conditions associated with hypotonia to be associated with short cord. (Therefore, the direction of the DAG may be the opposite for at least some of the NDD studied here (Fig 1), or cord length and NDD may share underlying paths.) Consistently, studies using animal models suggest that exposure to substances that decrease fetal movement is associated with short cords [28–30], and bone mineralization, which is promoted by physical activity, is lower in neonates with short cords, regardless of birthweight, sex, or placental weight [31].

Our finding that short cord was associated with impaired hearing is plausible, considering studies of the startle response to vibroacoustic stimulation. The startle response increases with gestational age, and there is evidence that male fetuses show larger responses, which again is in line with boys having longer cords from 28 weeks of gestation [12,32].

Male fetuses have higher energy requirements than females [33] and pregnant women carrying male fetuses have a higher energy intake [34]. This may support that male fetuses have a higher level of physical activity in the womb, longer cords, and thus males may be more vulnerable to cord compression and hypoxia. Experimental evidence in rodents suggests sex-differential mechanisms of protection against perinatal hypoxic–ischemic brain injury [35], which may contribute to the male vulnerability to develop NDD [22].

ASD showed a less consistent pattern of associations to cord length; a long cord was associated with ASD, but only in boys when stratified by sex. The associations we find between short cord and ID and epilepsy are in line with previous studies [20,21,36], but associations of cord length with CP, ADHD and impaired hearing are new.

Genetic and environmental factors contribute to the etiology of NDD [6,37,38], and both maternal and fetal factors influence the cord length [12]. Previously, we found that hypertension in pregnancy was positively associated with long cord, and negatively with short cord [12]. It has been disputed whether conditions associated with placental dysfunction such as hypertension, preeclampsia, and intrauterine growth restriction are associated with an increased risk of NDD, or if pregnancy related factors and NDD share genetic or environmental liability [39,40]. Unmeasured genetic confounding, i.e., when the same genetic factors are independently associated with both the exposure and outcome, has been shown to explain the observed associations between smoking during pregnancy and risk of ADHD in children [41,42]. Similarly, the same factors may be independently associated with cord length and NDD. However, our sensitivity analyses did not suggest that unmeasured confounding factors explain the relationship between cord length and NDD. The NDDs studied here may have a substantial genetic contribution to etiology [43,44] and a long and short umbilical cord show a risk of recurrence [12] suggesting that persisting environmental or genetic factors influence cord length. This putative relationship needs to be disentangled in later studies.

The modest strength of the associations observed in our study is expected since NDD has a multifactorial etiology, involving genetic, environmental, and epigenetic factors [45]. Furthermore, the genetic etiologies behind the NDD may overlap; ADHD-associated copy number and common genetic variants also show overlap with ones associated with schizophrenia, ASD and ID [43,44,46]. Also, there is a wide normal variation of cord length, and a variable time from the "exposure" during fetal life and the outcome (NDD diagnosis), and these conditions may be shaped by developmental trajectories. The association between cord length and NDD was less pronounced in preterm births. This may be due to a smaller size of the population, but also that (biologic) diversity increases with gestational age, making patterns less pronounced at a lower age. Moreover, preterm birth is a risk factor for NDD [38,47], and the cord length of those who are born preterm may not reflect the cord length of those still in the womb, making the interpretation of the associations between cord lengths and NDD complex in the preterm.

Our findings that especially ADHD, but also ASD, ID and epilepsy occurrence declined, suggest that a diagnosis of these conditions needs a longer observation, while impaired vision or hearing show almost no variation in occurrence by year of birth. Trends in the use of diagnostic tools, awareness and treatment may also influence the occurrences.

**Strengths and limitations**

The national population-based design, the prospective data collection, and the almost complete follow-up are the most important strengths of this study. Utilizing compulsory national registries eliminates recall and selection bias. This methodology has been employed in numerous previous studies [40,48].

There is a possibility of misclassification of the milder phenotypes of NDD. However, this misclassification is most likely non-differential. There may be differences in the phenotypes of NDD between the sexes, with ongoing debates on whether biases hinder diagnosis of, e.g., ASD and ADHD in girls [49,50]. Since cord length and fetal size are correlated and reduced fetal growth has been linked to increased risk of NDD/ADHD [51,52] we additionally conducted analyses using birthweight specific percentiles for umbilical cord length, although most of the associations found in the initial analyses remained. A recent mendelian randomisation analyses suggests that fetal growth is not causally associated with NDD [51]. However, the association between short cord and epilepsy was attenuated when birth weight specific percentiles were applied (S2 Table). It is also a strength that we included various categories of NDD in our study, making it possible to assess similarities or differences in the associations to cord length.

Many of the variables utilized here, including cord length have been validated [53,54], but not all variables in the MBRN have undergone validation. Studies have shown that the method of using NIS and NPR to identify persons with the NDD has high sensitivity and specificity [48,54]. The dose-response pattern shown for cord length percentile and ADHD, ID and impaired hearing is reassuring. It may be considered a limitation that we did not censor emigration or death, but if persons died or emigrated, they would not appear in the NIS or NPR. It may be reasonable to assume that families with affected children rarely emigrate.

As most NDD are heterogenous in nature, at clinical as well as etiological, and pathophysiological levels, a weakness of this study is that we did not have information on the phenotypes or severities of the studied conditions.

To address the limitation of observational studies regarding confounding, we carried out sensitivity analyses that showed that the likelihood of unmeasured confounding significantly affecting our results is small. Also, when including co-variates into the regression models the associations between cord length and NDD remained consistent, lending credibility to our results.

Whether the insights from our study may be useful to identify future children at risk is discussable, but our study generates hypotheses concerning early human development. Our research provides support to the notion that fetal physical activity is a determinant of the cord length, or at least that cord length may serve as a proxy for neurobehavioral activity and development and thus may represent an accessible measure to assure this.

## Conclusion

We provide prospective, population-based evidence that umbilical cord length is associated with NDD in a distinct pattern, supporting that neurodevelopment and development of the umbilical cord share pathways.

## Supporting information

**S1 Supplement. ICD-9 and 10 codes with the diagnoses.**
(DOCX)

**S1 Table. Bayesian sensitivity analysis.**
(DOCX)

**S2 Table. Associations between birth weight specific umbilical cord length percentiles and neuro developmental disorders.**
(DOCX)

## Author contributions

**Conceptualization:** Cathrine Ebbing, Anne Halmoy, Svein Rasmussen, Karen K. Mauland.

**Data curation:** Cathrine Ebbing, Svein Rasmussen, Dag Moster.

**Formal analysis:** Cathrine Ebbing, Svein Rasmussen.

**Funding acquisition:** Dag Moster.

**Investigation:** Cathrine Ebbing, Anne Halmoy, Svein Rasmussen.

**Methodology:** Cathrine Ebbing, Anne Halmoy, Svein Rasmussen, Karen K. Mauland, Dag Moster.

**Project administration:** Cathrine Ebbing.

**Supervision:** Svein Rasmussen, Dag Moster.

**Validation:** Anne Halmoy.

**Visualization:** Jørg Kessler.

**Writing – original draft:** Cathrine Ebbing, Anne Halmoy, Svein Rasmussen, Karen K. Mauland, Jørg Kessler.

**Writing – review & editing:** Cathrine Ebbing, Anne Halmoy, Svein Rasmussen, Karen K. Mauland, Jørg Kessler, Dag Moster.

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
