## [Decision Letter · Decision Letter 0]

10 Feb 2025

PONE-D-24-37689Umbilical cord length and neurodevelopmental disorders, a national cohort studyPLOS ONE

Dear Dr. Ebbing,

Thank you for submitting your manuscript to PLOS ONE. After careful consideration, we feel that it has merit but does not fully meet PLOS ONE’s publication criteria as it currently stands. Therefore, we invite you to submit a revised version of the manuscript that addresses the points raised during the review process.

We look forward to receiving your revised manuscript.

Kind regards,

Shivanand Kattimani

Academic Editor

PLOS ONE

Journal requirements:   When submitting your revision, we need you to address these additional requirements. 1. Please ensure that your manuscript meets PLOS ONE's style requirements, including those for file naming. The PLOS ONE style templates can be found at https://journals.plos.org/plosone/s/file?id=wjVg/PLOSOne_formatting_sample_main_body.pdf and https://journals.plos.org/plosone/s/file?id=ba62/PLOSOne_formatting_sample_title_authors_affiliations.pdf. 2. We note that the grant information you provided in the ‘Funding Information’ and ‘Financial Disclosure’ sections do not match.  When you resubmit, please ensure that you provide the correct grant numbers for the awards you received for your study in the ‘Funding Information’ section 3. Thank you for stating the following financial disclosure:  [The project was supported by the Gerda Meyer Nyquist Gulbrandson and Gerdt Meyer Nyquist’s Fund, and the Norwegian SIDS and Stillbirth Society https://lub.no/. Prof. Dag Moster was awarded the grants. The funders played no role in conducting the research and writing the paper].  Please state what role the funders took in the study.  If the funders had no role, please state: ""The funders had no role in study design, data collection and analysis, decision to publish, or preparation of the manuscript."" If this statement is not correct you must amend it as needed. Please include this amended Role of Funder statement in your cover letter; we will change the online submission form on your behalf. 4. Thank you for stating the following in the Acknowledgments Section of your manuscript: [The project was supported by the Gerda Meyer Nyquist Gulbrandson and Gerdt Meyer Nyquist’s Fund, and the Norwegian SIDS and Stillbirth Society. The funders played no role in conducting the research and writing the paper.]We note that you have provided funding information that is not currently declared in your Funding Statement. However, funding information should not appear in the Acknowledgments section or other areas of your manuscript. We will only publish funding information present in the Funding Statement section of the online submission form. Please remove any funding-related text from the manuscript and let us know how you would like to update your Funding Statement. Currently, your Funding Statement reads as follows:  [The project was supported by the Gerda Meyer Nyquist Gulbrandson and Gerdt Meyer Nyquist’s Fund, and the Norwegian SIDS and Stillbirth Society https://lub.no/. Prof. Dag Moster was awarded the grants. The funders played no role in conducting the research and writing the paper.] Please include your amended statements within your cover letter; we will change the online submission form on your behalf. 5. We note that you have indicated that there are restrictions to data sharing for this study. For studies involving human research participant data or other sensitive data, we encourage authors to share de-identified or anonymized data. However, when data cannot be publicly shared for ethical reasons, we allow authors to make their data sets available upon request. For information on unacceptable data access restrictions, please see http://journals.plos.org/plosone/s/data-availability#loc-unacceptable-data-access-restrictions.  Before we proceed with your manuscript, please address the following prompts: a) If there are ethical or legal restrictions on sharing a de-identified data set, please explain them in detail (e.g., data contain potentially identifying or sensitive patient information, data are owned by a third-party organization, etc.) and who has imposed them (e.g., a Research Ethics Committee or Institutional Review Board, etc.). Please also provide contact information for a data access committee, ethics committee, or other institutional body to which data requests may be sent. b) If there are no restrictions, please upload the minimal anonymized data set necessary to replicate your study findings to a stable, public repository and provide us with the relevant URLs, DOIs, or accession numbers. Please see http://www.bmj.com/content/340/bmj.c181.long for guidelines on how to de-identify and prepare clinical data for publication. For a list of recommended repositories, please see https://journals.plos.org/plosone/s/recommended-repositories. You also have the option of uploading the data as Supporting Information files, but we would recommend depositing data directly to a data repository if possible. Please update your Data Availability statement in the submission form accordingly. 6. Please include your full ethics statement in the ‘Methods’ section of your manuscript file. In your statement, please include the full name of the IRB or ethics committee who approved or waived your study, as well as whether or not you obtained informed written or verbal consent. If consent was waived for your study, please include this information in your statement as well.  7. Please include captions for your Supporting Information files at the end of your manuscript, and update any in-text citations to match accordingly. Please see our Supporting Information guidelines for more information: http://journals.plos.org/plosone/s/supporting-information. 

Additional Editor Comments:

We have now received reviewer comments.

Kindly revise and submit the manuscript within four weeks time.

If you want you request for extension.

Reviewers' comments:

Reviewer's Responses to Questions

**Comments to the Author**

1. Is the manuscript technically sound, and do the data support the conclusions?

Reviewer #1: Partly

Reviewer #2: Yes

2. Has the statistical analysis been performed appropriately and rigorously? 

Reviewer #1: I Don't Know

Reviewer #2: Yes

3. Have the authors made all data underlying the findings in their manuscript fully available?

Reviewer #1: No

Reviewer #2: Yes

4. Is the manuscript presented in an intelligible fashion and written in standard English?

Reviewer #1: Yes

Reviewer #2: Yes

5. Review Comments to the Author

Reviewer #1: The objectives of this paper has been discussed and explored as mentioned. Proper ethical clearance has been taken from concerned departments .Strength and limitations have been mentioned.

However in the methodology there is no clear definition of what is long and what is short cord. 108 "Exposures were short below 5th percentile and long above 95th percentile". This needs to have a reference to gestational age specific umbilical cord growth chart, as data regarding term and preterm and association with umbilical cord length and NDD have been presented in your study.

In the Covariates, maternal socioeconomic status has not been considered and discussed, though Figure 1 acyclic graph has included sociodemography as a covariate.

There was brief mention of the mothers' BMI and Ethnicity association with umbilical cord length. 205 "we repeated the analysis with NDD outcome restricted to cases with information on maternal BMI.We found no effect of maternal BMI on the risk estimate. This statement needs to be supported by data.

Figure 4 of data sheet needs to be clearly labelled

Reviewer #2: Very important study. calculation of sample size need clarification in methodology. Definition of short and long umbilical cord criteria or guideline and references is not addressed well in methodology. Avoid the abbreviations on abstract. The control of variances or standard competence process of midwife or physician who examine the neonate and measure the umbilical cord is not clear.

6. PLOS authors have the option to publish the peer review history of their article (what does this mean? ). If published, this will include your full peer review and any attached files.

**Do you want your identity to be public for this peer review?** For information about this choice, including consent withdrawal, please see our Privacy Policy .

Reviewer #1: **Yes: ** Shabina Ahmed

Reviewer #2: No

---

## [Author Response · Author response to Decision Letter 0]

3 Mar 2025

Response to review

PONE-D-24-37689

Dear Academic Editor Shivanand Kattimani

We thank you and referees for useful comments. Please find our responses to the comments itemised below.

[The project was supported by the Gerda Meyer Nyquist Gulbrandson and Gerdt Meyer Nyquist’s Fund, and the Norwegian SIDS and Stillbirth Society https://lub.no/. Prof. Dag Moster was awarded the grants. The funders played no role in conducting the research and writing the paper].

• Reply: this is now changed.: Should read:

“The project was supported by the Gerda Meyer Nyquist Gulbrandson and Gerdt Meyer Nyquist’s Fund, and the Norwegian SIDS and Stillbirth Society https://lub.no/. Prof. Dag Moster was awarded the grant with grant number 102230106. The funders had no role in study design, data collection and analysis, decision to publish, or preparation of the manuscript.”

…we will change the online submission form on your behalf.

• Thank you.

Reviewer: 1

The objectives of this paper has been discussed and explored as mentioned. Proper ethical clearance has been taken from concerned departments. Strength and limitations have been mentioned.

However in the methodology there is no clear definition of what is long and what is short cord. 108 "Exposures were short below 5th percentile and long above 95th percentile". This needs to have a reference to gestational age specific umbilical cord growth chart, as data regarding term and preterm and association with umbilical cord length and NDD have been presented in your study.

• Reply: We thank you for this comment. The percentiles of cord length were short (<5th empirical percentile) or long (≥95th empirical percentile) based on the cord lengths in the total study population. Empirical percentiles were constructed, based on gestational age at birth.

We have previously published reference ranges for cord length in PLOS one (reference no 12 in the current manuscript) 1. However, as the current study population is the same as in Linde et al 2018, we calculated and applied empirical percentiles based on the current study population. The method applied is well described in the reference paper (Linde, 2018). We have amended the text to make this clearer: The paragraph now reads:

“Exposures were short (<5th percentile) and long (≥95th percentile) umbilical cord. The cord length percentiles were empirical, i.e. constructed based on the distribution of umbilical cord length by parity, sex and gestational age in the study population. Since cord length increases with fetal size, we additionally constructed birthweight-specific percentiles for cord length. All data in the MBRN are registered prospectively. The attending midwife or physician conducted the examinations of the afterbirth and the neonate and entered the requested information into the registration form shortly after delivery. All neonates were examined by a physician and any malformation was noted. The umbilical cord was measured with a disposable tape and length reported in centimeters.”

In the Covariates, maternal socioeconomic status has not been considered and discussed, though Figure 1 acyclic graph has included sociodemographic as a covariate.

• Reply: Thank you. We had access to important socioeconomic variables, described in the “Covariates” section; ”maternal and paternal age and educational level, …marital status of the mother… maternal and paternal level of education (categorized into five levels) and immigrant status…”. We have added a footnote to Figure 1 Directed acyclic graph: “Social demographic factors contain maternal and paternal education level, marital status of the mother, immigration status of the child.”

There was brief mention of the mothers' BMI and Ethnicity association with umbilical cord length. 205 "we repeated the analysis with NDD outcome restricted to cases with information on maternal BMI. We found no effect of maternal BMI on the risk estimate. This statement needs to be supported by data.

• Reply: We carried out analyses in the selection of the population where BMI was available. Including BMI in the model did not change the results; for the association between a long cord and ADHD OR in this subset was 1.07 (95%CI 0.93-1.24) and including BMI in the model aOR 1.02 (95%CI 0.88-1.17) and short cord and ID; OR 2.67 (95%CI 2.06-3.47) including BMI in the model; aOR 2.75 (95%CI 2,12-3.58). Please see the results in the table below. At the discretion of the Editor, we may add this as supplemental material, although we feel this may be too much information. The paragraph now reads:

“Since maternal BMI before pregnancy was available only in a subset of the population , we repeated the analyses (with NDD as outcomes) restricted to cases with information on maternal BMI (n=174,692). We found no effect of adjusting for maternal BMI on the risk estimates. Likewise, no effect was observed on the risk estimates when we adjusted for ART.”

Population with 174692 BMI data:

Exposure Outcome OR 95%CI aOR with BMI 95%CI

long cord ADHD 1.07 0.93 1.24 1.02 0.88 1.17

short cord ID 2.67 2.06 3.47 2.75 2.12 3.58

short cord Impaired hearing 1.42 1.17 1.74 1.45 1.19 1.77

short cord Epilepsy 1.48 1.16 1.89 1.51 1.18 1.92

short cord CP 1.47 0.91 2.37 1.52 0.94 2.44

long cord CP 1.18 0.75 1.86 1.12 0.71 1.76

Figure 4 of data sheet needs to be clearly labelled

• Reply: We have now inserted the requested information in the Figure text.

Figure text now reads:

“Figure 1 Directed acyclic graph (DAG) describing the relationship between pregnancy, maternal and paternal factors, umbilical cord length and neurodevelopmental disorder in the child. White nodules represent covariates that are included in the model. Social demographic factors contain maternal and paternal education level, marital status of the mother, immigration status of the child. Grey nodule; Unknown confounding factors, e.g. genetic confounding.

Figure 2 The prevalence of neurodevelopmental disorders according to year of birth in Norway

Figure 3 The prevalence of neurodevelopmental disorders according to cord length percentile

Figure 4 Long or short umbilical cord (≥95th or <5th percentile) and risk of neurodevelopmental disorders among singletons. Panel A; exposure Long Cord, panel B; exposure Short Cord.

CP; Cerebral Palsy, ADHD; Attention-deficit/hyperactivity disorder, ASD; Autism Spectrum Disorder; ID; Intellectual Disability, OR; Odds Ratio, CI; Confidence Interval”

Reviewer #2: Very important study.

• Reply: Thank you.

• calculation of sample size need clarification in methodology.

• Reply: This is an observational population study, including all births in Norway during the study period, thus a sample size calculation is not relevant in this setting.

Definition of short and long umbilical cord criteria or guideline and references is not addressed well in methodology.

• Reply: Thank you! Please see reply to Reviewer no1 above.

Avoid the abbreviations on abstract.

• Reply; this is difficult due to word limit. The guide for authors states that abbreviations should be avoided ”if possible”. In our situation both ADHD and ASD should be easier to read than the full description.

The control of variances or standard competence process of midwife or physician who examine the neonate and measure the umbilical cord is not clear.

• Reply: We have included a reference in the discussion section regarding the strengths and limitations (ref no 53 in the manuscript and ref 2 here below). This study was a validation study of the classification and measurements of the placenta, membranes and cord carried out by midwives in a clinical setting. The inter- and intra-observer study showed no significant differences regarding placental weight and cord length, indicating that the validity of data regarding placenta and cord in the Norwegian birth register is sufficiently high to justify future large-scale epidemiologic research based on this database.

•

1. Linde LE, Rasmussen S, Kessler J, Ebbing C. Extreme umbilical cord lengths, cord knot and entanglement: Risk factors and risk of adverse outcomes, a population-based study. PloS one. 2018;13(3):e0194814. doi:10.1371/journal.pone.0194814

2. Sunde ID, Vekseth C, Rasmussen S, Mahjoob E, Collett K, Ebbing C. Placenta, cord and membranes: a dual center validation study of midwives' classifications and notifications to the Medical Birth Registry of Norway. Acta Obstet Gynecol Scand. Sep 2017;96(9):1120-1127. doi:10.1111/aogs.13164

---

## [Decision Letter · Decision Letter 1]

23 Mar 2025

Umbilical cord length and neurodevelopmental disorders, a national cohort study

PONE-D-24-37689R1

Dear Dr. Ebbing,

We’re pleased to inform you that your manuscript has been judged scientifically suitable for publication and will be formally accepted for publication once it meets all outstanding technical requirements.

Kind regards,

Shivanand Kattimani

Academic Editor

PLOS ONE

Reviewers' comments:

Reviewer's Responses to Questions

**Comments to the Author**

1. If the authors have adequately addressed your comments raised in a previous round of review and you feel that this manuscript is now acceptable for publication, you may indicate that here to bypass the “Comments to the Author” section, enter your conflict of interest statement in the “Confidential to Editor” section, and submit your "Accept" recommendation.

Reviewer #1: All comments have been addressed

Reviewer #2: All comments have been addressed

2. Is the manuscript technically sound, and do the data support the conclusions?

Reviewer #1: Yes

Reviewer #2: Yes

3. Has the statistical analysis been performed appropriately and rigorously? 

Reviewer #1: Yes

Reviewer #2: Yes

4. Have the authors made all data underlying the findings in their manuscript fully available?

Reviewer #1: Yes

Reviewer #2: Yes

5. Is the manuscript presented in an intelligible fashion and written in standard English?

Reviewer #1: Yes

Reviewer #2: Yes

6. Review Comments to the Author

Reviewer #1: (No Response)

Reviewer #2: The article is good and presented an important clinical outcome. There is no ethical concern, the article is designed well, address important aim and conclusion

7. PLOS authors have the option to publish the peer review history of their article (what does this mean? ). If published, this will include your full peer review and any attached files.

**Do you want your identity to be public for this peer review?** For information about this choice, including consent withdrawal, please see our Privacy Policy .

Reviewer #1: **Yes: ** Shabina Ahmed

Reviewer #2: No

---

## [Editor Report · Acceptance letter]

PONE-D-24-37689R1

PLOS ONE

Dear Dr. Ebbing,

I'm pleased to inform you that your manuscript has been deemed suitable for publication in PLOS ONE. Congratulations! Your manuscript is now being handed over to our production team.

Kind regards,

on behalf of

Dr. PLOS Manuscript Reassignment

Staff Editor

PLOS ONE